# Biomass Allocation to Resource Acquisition Compartments Is Affected by Tree Density Manipulation in European Beech after Three Decades

**Bohdan Konôpka [1,2], Milan Barna [3,\*], Michal Bosela [4] and Martin Lukac [2,5]**

[1]  Forest Research Institute, National Forest Centre, T.G. Masaryka 22, SK-960 01 Zvolen, Slovakia; bohdan.konopka@nlcsk.org

[2]  Faculty of Forestry and Wood Sciences, Czech University of Life Science Prague, Kamýcká 129, CZ-165 00 Praha 6–Suchdol, Czech Republic; m.lukac@reading.ac.uk

[3]  Institute of Forest Ecology, Slovak Academy of Science, Ľ. Štúra 2, SK-960 53 Zvolen, Slovakia

[4]  Faculty of Forestry, Technical University in Zvolen, T.G. Masaryka 24, SK-960 01 Zvolen, Slovakia; michal.bosela@tuzvo.sk

[5]  School of Agriculture, Policy and Development, University of Reading, Reading RG6 6EU, UK

\*  Correspondence: barna@ife.sk

**Abstract:** This study reports on an investigation of fine root and foliage productivity in forest stands dominated by European beech (*Fagus sylvatica* L.) and exposed to contrasting intensities of mature forest harvesting. The main aim of this study was to consider the long-term effects of canopy manipulation on resource acquisition biomass compartments in beech. We made use of an experiment established in 1989, when five different light availability treatments were started in plots within a uniform forest stand, ranging from no reduction in tree density to full mature forest removal. We measured fine root standing stock in the 0–30 cm soil layer by coring in 2013 and then followed annual fine root production (in-growth cores) and foliage production (litter baskets) in 2013–2015. We found that the plot where the tree density was reduced by 30% had the lowest foliage and the highest fine root production. In 2013, this plot had the highest fine root turnover rate (0.8 year$^{-1}$), while this indicator of fine root dynamics was much lower in the other four treatments (around 0.3 year$^{-1}$). We also found that the annual fine root production represented around two thirds of annual foliage growth on the mass basis in all treatments. While our findings support the maintenance of source and sink balance in woody plants, we also found a long-lasting effect of tree density manipulation on investment into resource acquisition compartments in beech forests.

**Keywords:** *Fagus sylvatica;* fine roots; foliage; standing stock and production; root-to-shoot ratio

## 1. Introduction

European beech (*Fagus sylvatica* L., beech hereafter) is one of the most important tree species in Europe—both commercially and ecologically. The current distribution range of beech extends from Southern Scandinavia to the south of Italy, and from Spain to Northeast Turkey [1]. In Slovakia, beech is the most common tree species and covers around 34% of the country's forested area [2]. Here, beech stands occur within an altitudinal range spanning from 250 m above sea level (a.s.l) up to 1250 m a.s.l. [3]. Beech represents not only the most important species for commercial forestry but is also extremely important ecologically since it is relatively resistant to ongoing climate change [4,5], notably in mixed-species forests [6].

European beech is tolerant of shade conditions, especially during the initial stages of growth when shielding by a mature canopy aids regeneration [1]. The shelter-wood silviculture system and its variations, thus, appear to be the most suitable for beech regeneration in commercial forests [3]. The degree of canopy opening in mature stands has been shown to be the main factor affecting the abundance and growth rate of beech seedlings and saplings [7]. Conditions below the main canopy influence not only the speed of growth but also affect biomass allocation to individual compartments in young beech trees [8]. Light availably is known to drive carbohydrate allocation to acquisition compartments, such as foliage and fine roots, and to alter their contribution to whole tree biomass. Following a harvesting operation in a closed canopy forest, seedlings and saplings experience an immediate change in aboveground resource availability [9]. In time, this change may be followed by an increase in water and nutrient availabilities in the soil as the root systems of harvested trees cease to function and start to decompose [10]. The key difference between the dynamics of light and soil resource availabilities over time is that changes in available light are robust and changes in soil resources may be subtle [11].

The life span of an individual plant tissue is a reflection of its utility to the organism as a whole. For example, while the lifespans of the stem, coarse branches and coarse roots usually correspond to that of the age of the tree, while the lifespans of the leaves and fine roots range between a couple of months to several years. [12]. Foliage and fine roots are ephemeral; their purpose is the acquisition of resources from the environment. As soon as their cost-effectiveness erodes, they senesce and are abscised and replaced [13]. Since a substantial portion of tree biomass is composed of carbon, the lifespan of tree compartments is a determinant of carbon retention time. Moreover, as light availability to beech trees determines carbon allocation to biomass compartments with vastly different turnover rates, it may act as a driver of carbon retention within the forest ecosystem.

The quantification of foliage biomass stock is relatively straightforward, either based on destructive (tree harvest and separation of leaves [14]) or non-destructive methodologies (litterfall capture—see, for instance, chapter 13 in ICP Forests Manual [15]). On the other hand, methods of measuring fine root biomass stock and production are more difficult, time-consuming, and less precise [16]. For this reason, studies analysing foliage production and turnover are far more frequent in the literature than those looking at the corresponding belowground acquisition biomass compartment. Furthermore, most information originates from observations gathered immediately after an intervention, not long-term observations. As a result, our understanding of the carbon investment into foliage and fine roots is still insufficient and merits closer attention. For example, we need to consolidate their considerable contribution of short-rotating tree components to net primary production [17,18].

The main objective of this paper is to quantify foliage and fine root productivity and to evaluate the effect of light availability on carbon allocation to these two biomass compartments, decades after changing growth conditions in a stand. We made use of a tree density manipulation experiment established in 1989, with the goal of observing the long-term effects of natural regeneration and subsequent beech forest growth on its resource capture capacity. Specifically, we were interested in whether, three decades after forest management interventions, there was a discernible effect on: (i) fine root biomass and production, (ii) depth allocation of fine root biomass and (iii) the relationship between above- and belowground annual production of acquisition organs.

## 2. Material and Methods

### 2.1. Site and Stand Conditions

The Ecological Experimental Site (EES) is situated about 6 km northwest from Kováčová village, Kremnické vrchy Mountains, Western Carpathians, Slovakia (48°38′ N and 19°04′ E). This area is characterised by its volcanic origin, andesite bedrock, and stony andosol soil. The stone content is 30–40% on average and increases with soil depth [19]. The research site is situated on a west-oriented aspect with 20% slope and altitude range between 450 and 510 m a.s.l. The 30 year mean annual temperature is 6.8 °C, and that of the growing season, which typically lasts from April to September,

is 13.5 °C. The mean annual precipitation is just under 800 mm, with about 450 mm in the growing season. The dominant vegetation association of EES is *Dentario bulbiferae-Fagetum*, with some incursions of the association *Carici pilosae-Fagetum* [20]. The vegetation cover consists mostly of permanent elements, such as *Carex pilosa* Scop., *Dentaria bulbifera* L., *Galium odoratum* Scop., as well as *Athyrium filix-femina* L. (Roth) and *Dryopteris filix-mas* (L.) Schott.

## 2.2. Plot and Subplot Design

Starting in the winter 1988/1989, an approximately 90 years old beech-dominated stand, composed of beech—62%, fir—22%, oak—7%, hornbeam—6%, lime—3%, was divided into five adjacent plots (Figure 1). The size of the whole forest stand was about 4.5 ha, while each plot covered an area of 0.35 ha. Three plots were subjected to shelterwood cutting with varying intensity—one was clear cut and one was left with no intervention as a control. In the shelterwood system, the mature stand is usually removed in a series of two to four cuts. The early cuts are designed to progressively improve vigour and seed production of the remaining trees while improving the conditions for new seedlings. The final cut is made when a sufficient amount of desirable regeneration has been achieved and when the shelter becomes a hindrance to the growth of the seedlings, rather than a benefit, it is necessary to remove the remainder of the mature stand [21]. The aim was to generate stands with different stocking density within each plot and, at the same time, serving as models of the phases of the shelterwood management system. Four plots were subjected to progressive cutting, graded as follows—LC plot (light cut), MC plot (medium cut), HC plot (heavy cut), CC plot (clear cut)—whilst the fifth plot was left without intervention—NC plot (no cut). The early cuts were primarily focused on the dying and damaged trees, trees of very low quality, admixed species (fir (*Abies alba* Mill.), lime (*Tilia cordata* Mill), hornbeam (*Carpinus betulus* L.)). After the initial harvest, the stands were composed of 160–700 mature trees per hectare, with an average stand height of 23.6–27.7 m and basal area of 13.5–40.9 m² (see [22,23] for more details). Prior to this research, the plots were managed uniformly and according to the local forestry practice. A second cutting was performed in 2004, focusing on the removal of all remaining trees within the HC plot and a reduction in relative density of 30 in MC and 50 in LC plots. Finally, in 2009, the removal of all original trees was conducted in the MC and LC plots (Table 1). The relative density was calculated as the ratio between the observed basal area of plots and the maximum or potential basal area defined by yield models [24,25,6]. In 2013, three 3 × 3 m subplots were established within each of the five plots. The subplots were established randomly about 20 m from each other (Figure 1).

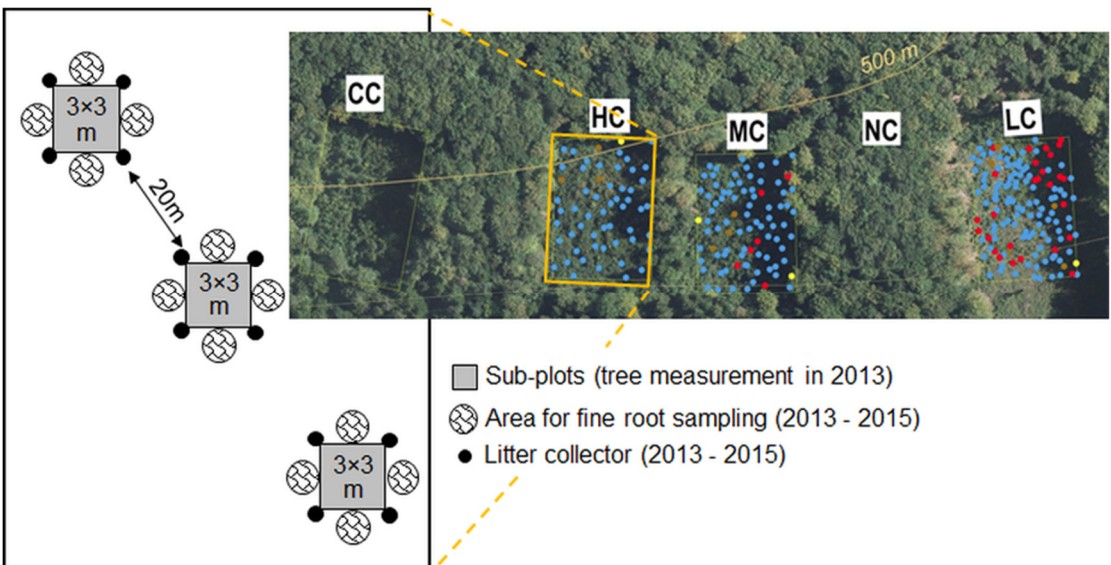

**Figure 1.** Sampling scheme established within each plot (treatment). Three subplots at least 20 m distant were used for tree measurements, accompanied by twelve sampling locations for fine root and foliage quantification (*n* = 3). The aerial view inset shows the experimental site in 2009. There are five

plots (left to right: CC (clear cut), HC (heavy cut), MC (medium cut), NC (control plot—no cut), LC (light cut)). Coloured dots indicate the locations of adult trees left after the first shelterwood cut in 1989: beech (bleu), fir (red), hornbeam (yellow) and oak (brown).

**Table 1.** Timing and intensity of forest management operations in experimental forest plots, all interventions were carried out in February–March prior to the onset of the vegetation season. All operations relate to the mature stand only (red.BA—% reduction in original basal area, BA—basal area after intervention (m² ha⁻¹), RD—relative density calculated as actual over theoretical full (yield tables) basal area (%), NC—no cut (control plot), LC—light cut, MC—medium cut, HC—heavy cut, CC—clear cut).

| Plot | Silvicultural System | First Harvest (1989) | | | Second Harvest (2004) | | | Third Harvest (2009) | | |
|------|----------------------|---------|-----|-----|---------|-----|-----|---------|-----|-----|
| | | Red. BA | BA | RD | Red. BA | BA | RD | Red. BA | BA | RD |
| NC | control plot (no cut) | 0 | 41 | 90 | 0 | 45 | 100 | 0 | 49 | 100 |
| LC | shelterwood (light cut) | 24 | 29 | 70 | 43 | 21 | 50 | 100 | 0 | 0 |
| MC | Shelterwood (medium cut) | 44 | 19 | 50 | 53 | 13 | 30 | 100 | 0 | 0 |
| HC | shelterwood (heavy cut) | 68 | 14 | 30 | 100 | 0 | 0 | 0 | | 0 |
| CC | clear cut | 100 | | 0 | 0 | | 0 | 0 | | 0 |

## 2.3. Tree Measurements

All young tree measurements reported in this study were conducted in August within the 9 m² replicate subplots established in each plot; natural regeneration tree number, species composition, tree height and diameter at breast height (DBH) and basal area were measured. A digital caliper was used to measure the DBH of young trees (at 1.3 m from ground level) to the nearest tenth of a millimetre. All tree measurements took place after the cessation of the girth expansion of all species [26]. The DBH of all trees taller than 1.3 m within a subplot was used to calculate the basal area of this stand. Trees smaller than 1.3 m were not measured for DBH and are reported as the number of trees only (stems ha⁻¹, Table 2). The regeneration of species other than beech was present in all subplots (hornbeam, fir, lime), but all were dominated by beech (40–88%) (see Table 2 and [27] for more details).

**Table 2.** Main forest stand characteristics in each experimental plot in 2013. Height—mean tree height with 25–75%—range between 25th and 75th percentile, beech—% proportion of beech.

| Plot | Silvicultural System | Growth Stage | Density (stems ha⁻¹) | Height (m) | | Beech |
|------|----------------------|--------------|----------------------|------------|--------|-------|
| | | | | Mean | 25–75% | (%) |
| NC | control plot (no cut) | mature | 600 | 29.1 | 21.3–32.5 | 94.4 |
| | | seedling | 42,600 | 0.3 | 0.1–0.3 | 79.3 |
| LC | shelterwood (light cut) | thicket | 41,900 | 2.2 | 1.4–3.0 | 77.3 |
| MC | shelterwood (medium cut) | thicket | 58,100 | 2.9 | 1.0–3.9 | 87.7 |
| HC | shelterwood (heavy cut) | thicket | 22,200 | 5.1 | 3.0–7.2 | 60.7 |
| CC | clear cut | thicket | 11,100 | 7.5 | 4.0–10.3 | 40.3 |

## 2.4. Fine Root Sampling and Processing

Fine root biomass and production were estimated by two methods: soil coring and in-growth cores [28]. Within each subplot, we extracted 4 soil cores in the spring of 2013 to measure standing fine root biomass (Figure 1). A metal auger with an inner diameter of 6.5 cm was used to extract soil core to a depth of 30 cm. The boundary between the litter and humus layers was considered zero soil depth. The core was divided into three 10 cm layers, representing soil depths 0–10, 10–20 and 20–30 cm. Each subsample was inserted into a labelled plastic bag transported to the laboratory and kept frozen at −20 °C until further processing.

In-growth cores were constructed from 2.5 mm polyamide mesh to fit the inside of a metal auger with a 6.5 cm inside diameter and 30 cm length. A soil opening was made with the abovementioned corer and a plastic net was inserted flush with the soil surface and then filled with compacted quartz sand. Four in-growth cores per subplot were installed at every occasion (60 in-growth cores per

season). Installations took place in mid-March of 2013, 2014 and 2015 and a new core was installed at least 50 cm from the previous installation. Used soil openings were backfilled when no longer needed. The cores were harvested at the end of each growing season, in early November 2013, 2014 and 2015. We assumed negligible fine root production during the dormant season and fine root turnover during the in-growth core exposition time. After extraction, in-growth cores were stored, as above.

Prior to processing, standing biomass and in-growth core samples were defrosted and fine roots (defined as those with a diameter up to 2 mm) were manually picked from the soil or the sand, first from free soil and then by wet sieving. Live tree fine roots were separated from dead on the basis of colour and vigour. Dead roots found in sequential cores were discarded as live root biomass was the focus of this study. We found only negligible amounts of dead roots in the in-growth cores. Live roots were separated into those from trees and other plant types (mostly grasses and herbs). The sorting was done visually due to very different colour and morphological structures of non-tree roots (e.g., fine tree roots were ramified with many tips, but ground layer plants had simple fibrous roots with nearly no ramification and few tips). Separated fine roots were carefully washed to remove adhering soil particles, oven-dried for 24 h under 75 °C in paper bags and weighed.

### 2.5. Foliage Sampling and Processing

Plastic flowerpots with an inner diameter of 30 cm and a height of 40 cm were placed on the soil surface and used to collect foliage litter. In March 2013, four collectors were placed within each subplot. They collected canopy litter continuously, describing annual litter production in 2013, 2014, and 2015. Litter was collected twice a month to minimize the loss of material from the collectors. Upon collection, all litter was placed into paper bags and stored in a dry well-ventilated room. At the end of the collection period in 2015, all litter was visually inspected and all components (buds, flowers, beech masts, small twigs, insects, etc.) except foliage were discarded. All samples were oven-dried for 48 h at 95 °C and weighed.

### 2.6. Calculations and Statistical Analyses

Foliage and fine root production were measured and expressed on annual bases, covering one growing season. Fine root turnover in 2013 was calculated as the ratio of fine root production as measured by the sand filled in-growth cores and standing fine root biomass. We used linear mixed-effects models to evaluate the effects of different management on the stock and production of foliage and fine roots. The mixed-effect model was adjusted to take into account the fact that experimental sub-plots are nested within harvest intensity plots, which were not replicated. The setting of random effects in the models varied depending on the variable tested. In the case of foliage and fine root production, treatment was used as a fixed factor, and year nested within the subplot was used as a random effect. This setting allowed us to consider change over time within each subplot (and avoid temporal autocorrelation if the year was used as a fixed factor). Year was dropped for the analysis of foliage and fine root stock in 2013; however, the model setting remained the same otherwise. A regression model with least-square method of the parameter estimation was fitted to the data to assess the influence of stand density and composition on biomass production.

All models were fitted using the restricted maximum likelihood by the "lmer" function [29] in R software [30]. The significance of model parameters was assessed via the Satterthwaite's method [31]. We used QQ plots to inspect if the residuals of the mixed models follow the normal distribution. Tukey post-hoc test was used to carry out pairwise comparisons of individual treatments. The function "emmeans" [32] in R was used to compute estimated marginal means of the treatment levels in the "lmer" models. The "pairs" function in R [33] was then used to compute the Tukey-adjusted *p*-values to quantify the significance of the pairwise differences between the treatments. Finally, "ggplot2" package in R was used to generate the figures [34].

### 3. Results

Forest management interventions initiated a contrasting development between the plots. Generally, the more intensive the initial harvest of trees, the faster the regeneration and young tree growth (Table 2). As expected, in 2013 the tallest regenerating trees were recorded in the CC plot (mean height 7.5 m), followed by HC (5.1 m), MC (3.0 m), and the LC plot (2.3 m). Tree density followed the reverse trend, the least number of trees was found in the NC plot (approx. 11,000 ha⁻¹), while the highest number of trees was recorded in the MC plot (58,000 ha⁻¹). The basal area of the newly regenerating forest in 2013, 24 years after the initial reduction in stand density, was the highest in the CC plot (50.1 m² ha⁻¹) and the lowest in the LC plot (7.1 m² ha⁻¹). The management of relative density changed tree species composition. At the time of observation, the largest proportion of beech was recorded in the NC plot (94.4% if considering mature trees), while the lowest contribution of beech was in the CC stand (40.3%).

In 2013, the standing stock of fine roots in the 0–30 cm mineral soil layer varied significantly between the plots, from 2.30 t ha⁻¹ in the LC to 5.14 t ha⁻¹ in the NC plot ($p = 0.004$; Figure 2A). The highest proportion of standing fine root biomass was found in the shallowest soil layer (0–10 cm) and the lowest in the deepest soil layer (20–30 cm) in all stand density treatments. Interestingly, the largest share of fine roots in the shallow soil (59.8%) was found in the NC plot and the lowest (45.6%) in the LC plot. The observed variation in fine root production was clearly driven by stand density treatments ($p < 0.001$), rather than an inter-annual variation in environmental conditions ($p = 0.801$). Looking at the three-year mean, the largest fine root annual production was recorded in the LC plot (2.07 t ha⁻¹) and the lowest in the NC plot (0.87 t ha⁻¹). Fine root turnover in 2013 ranged between 0.16 year⁻¹ in the NC plot and 0.8 year⁻¹ in the LC plot (Figure 2B).

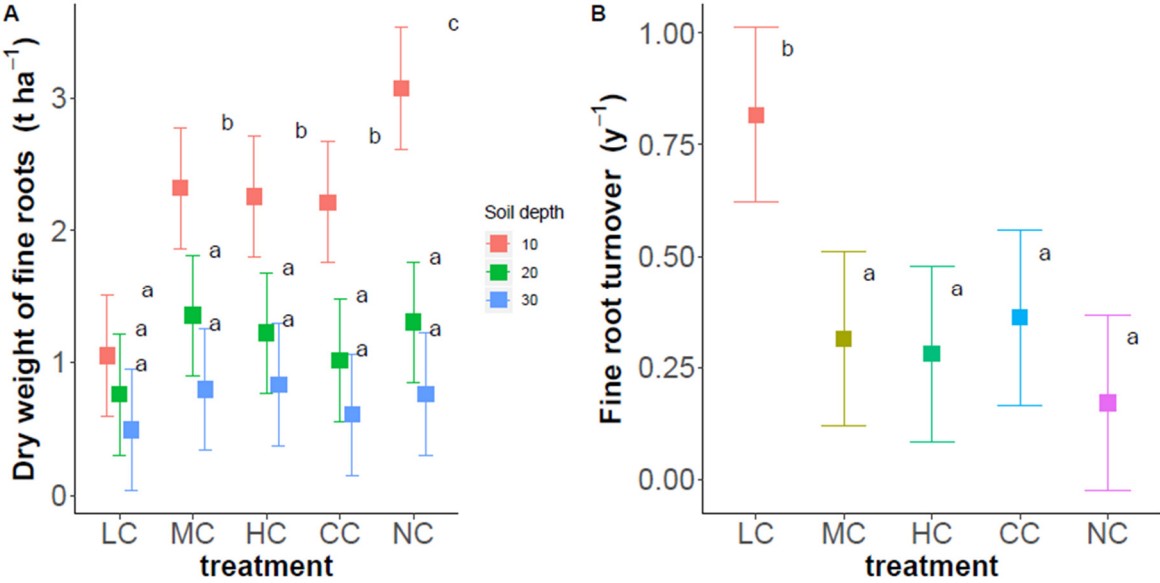

**Figure 2.** Standing stock of fine roots (**A**) and their turnover rate (**B**) in beech plots (using in-growth cores; mean of three sub-plots) with contrasting canopy density, 24 years after conversion from dense forest, specifically: clear-cut (CC, 100% relative density reduction), heavy cut (HC, 70%), medium cut (MC, 50%), light cut (LC, 30%) and no cut (NC, 0%). Soil depth represents mineral soil 0–10 cm, 10–20 cm and 20–30 cm depth. Letters indicate significant differences at $p < 0.05$; error bars show 95% confidence intervals.

The highest annual foliage production was observed in the NC plot, where 3.33 t ha⁻¹ year⁻¹ was produced annually on average. In contrast, the lowest amount of foliage was recorded in the LC plot (2.24 t ha⁻¹; Figure 3). Overall, annual foliage production increased significantly from 2013 (2.57 t ha⁻¹), through from 2014 (2.78 t ha⁻¹) to 2015 (3.03 t ha⁻¹; $p = 0.002$). The relationship between the annual production of foliage and fine roots appears to be stable over the period of observation ($p = 0.259$), while the root/shoot ratio in the NC treatment was significantly lower than in all other treatments ($p < 0.001$). Plotting the amount of foliage produced in 2013 as a function of standing fine

root biomass reveals a linear relationship across the range of density manipulation interventions studied here (Figure 4). Finally, fitting a simple linear regression model showed that there was a detectable effect of original stand density on the production of beech fine roots in the regenerated stands only ($p = 0.002$; $R^2 = 0.23$). However, the low number of replicate observations ($n = 15$) and their high variability do not allow for the construction of a meaningful multiple regression model.

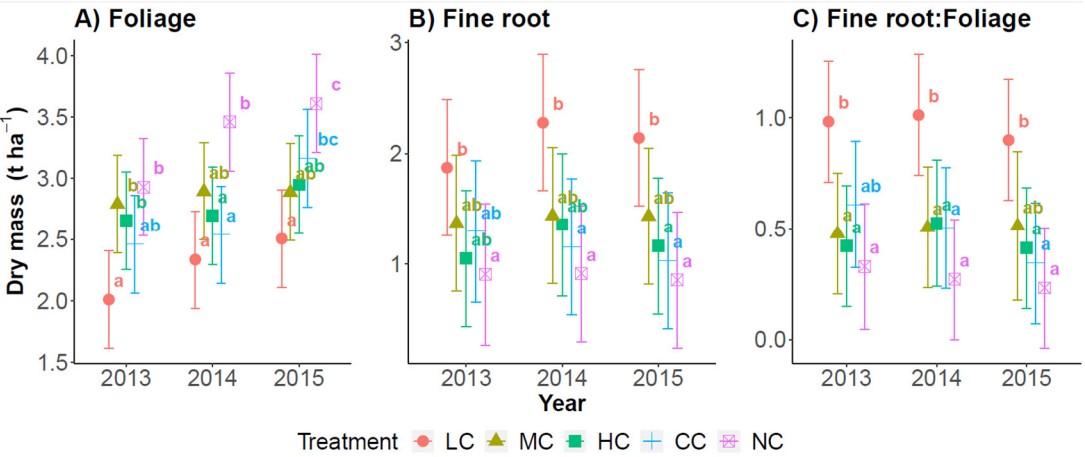

**Figure 3.** Annual foliage (**A**), fine root production (**B**) and fine root to foliage ratio (**C**) in beech plots with contrasting canopy densities in 2013, 2014 and 2015. Original stand density was reduced in 1989, specifically: clear-cut (CC, 100% relative density reduction), heavy cut (HC, 70%), medium cut (MC, 50%), light cut (LC, 30%) and no cut (NC, 0%). Letters indicate significant differences at $p < 0.05$; error bars show 95% confidence intervals.

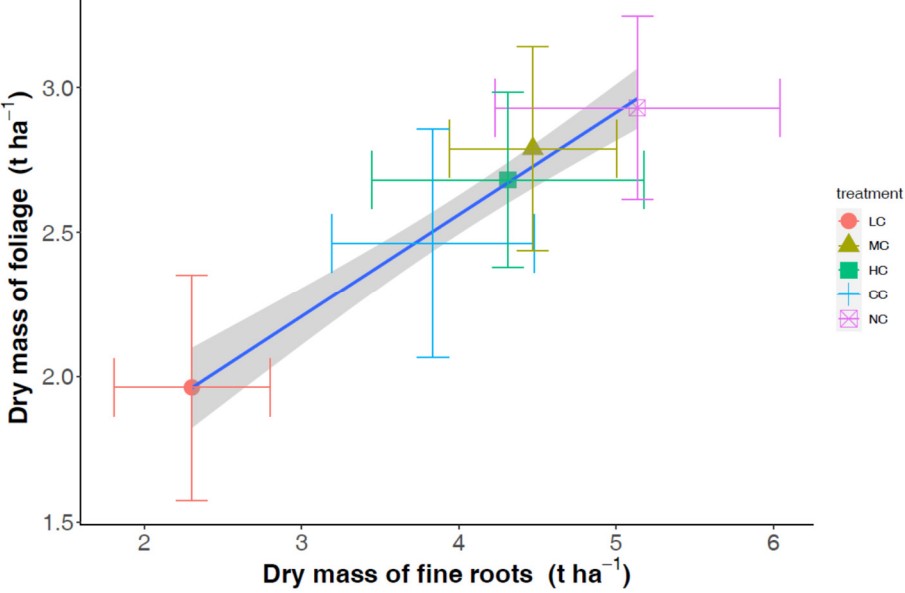

**Figure 4.** Relationship between standing stock of fine roots and foliage produced in five beech forest plots with contrasting tree densities in 2013. Original stand density was reduced in 1989, specifically: clear-cut (CC, 100% relative density reduction), heavy cut (HC, 70%), medium cut (MC, 50%), light cut (LC, 30%) and no cut (NC, 0%). Error bars show 95% confidence intervals point estimation. Grey area indicates 95% confidence interval of the regression line.

## 4. Discussion

The results from five forest plots dominated by European beech show that, nearly three decades after a reduction in tree density, the effects of increased light and soil resource availability are still clearly visible. One observation that stands out is that the treatment with the smallest reduction in

stand density (LC) is clearly distinct from all others. A possible explanation of this finding is the relatively short time period since the last intervention in this plot, when approximately 80% of light interception by the canopy was removed in 2009. The results also show different vertical distribution of fine roots, the LC treatment had the lowest, whereas the control NC treatment had the highest proportion of fine roots in the topsoil. The LC plot was overgrown with ground vegetation (see also [20]) to a much larger degree compared to the other four plots (data not shown). Virtually all fine roots of ground vegetation were found in the 0–10 cm layer, possibly leading to strong competitive pressure pushing fine beech roots to deeper soil.

The NC plot, interestingly, was significantly drier than the other treatments throughout the observation period (data not shown). Drier soil did not translate to lower fine root mass or production in this plot, despite numerous studies showing a positive relationship between soil moisture and root growth [35] with a second-order impact on nutrient availability [36]. The light cut plot was different not only in terms of allocation of carbon to foliage and fine roots but also fine root turnover. Clearly, the low reduction in tree density at the beginning, followed by a much later removal of tall forest, may have resulted in different stand development dynamics [37]. Stand age and stand productivity have a direct impact on the quantity of foliage and fine root biomass [38]. Alongside tree size, Tateno et al. [39] showed significant variability in net primary production allocation to tree compartments as a result of the variation in soil moisture and nitrogen supply. Finér et al. [40], in their analysis of a large root database, show that sites with more fertile soils typically result in smaller fine root standing stock in beech forests.

Being deciduous, the annual production of foliage in beech forests equals the amount of leaf litter—the turnover of this biomass compartment is thus equal to 1. Fine roots, however, do not conform to this pattern. Their turnover is much more difficult to establish and interpret. In this study, fine root turnover found in four treatments was around 0.3 year$^{-1}$, the only outlier was the light cut treatment where we saw a higher rate of turnover of 0.8 year$^{-1}$. The latter confirms existing observations—mean fine root turnover in beech forests in Europe was estimated as 0.86 year$^{-1}$ [12], while the general figure for all European forests is thought to be near 0.75 year$^{-1}$ [41]. The values observed in the NC, HC, MC, and CC are thus fairly low. Nevertheless, since the annual fine root production observed in this study represents around two thirds of annual foliage growth on a mass basis in all treatments, the low turnover rates are correspondent with a slow build-up of fine root mass present in the soil at any point in time. The use of ingrowth cores filled with sand introduced several artefacts inherent to this method. A direct comparison with fine root turnover rates measures elsewhere should be done with caution, but our data allow for an assessment of the differences between the treatments in this study as all were affected by the same experimental error. However, fine root turnover values in this study correspond to annual foliage production, which can be measured with much greater accuracy. Thus, it appears that light availability does not affect the ratio of carbon investment into above- and belowground acquisitive compartment, lending evidence to the theory describing the conservation of source and sink balance in plants [42].

The regeneration of beech was initiated by the reduction in tall forest tree density in 1989—the larger the reduction, the smaller the representation of beech. The subsequent interventions did not have a significant influence on the proportion of beech in natural regeneration [27]. The abundance of natural regeneration is driven by seed availability, sprouting success, survival and growth [43–45]. Seedling germination likely does not depend on light availability, but seedling survival and successive growth is highly dependent on it [45]. In this study, the most abundant fruiting beech trees were found in the densest stands (NC and LC). Further, these two stands had the lowest light availability close to the ground: 2–4% of open area light in NC and 8% in LC (Table 1). As beech is a very strong competitor in low-light conditions, leaving the stands to self-development would result in almost complete beech dominance [46]. Should forest management aim to deliver benefits other than productivity—for example, higher tree species diversity—different intensities of interventions are needed to support the regeneration and survival of other species [47]. In fact, the only plot with tree species diversity different from that of the NC plot was the CC plot, indicating that forest management interventions can influence the structure and diversity of regeneration [27].

There are several limitations of our study which must be considered and possibly addressed by future research. For instance, the very high portion of stones in the soil under our plots means that we were not able to sample roots in deeper soil layers, limiting our information to the top 30 cm of mineral soil. In general, most of the fine root biomass is usually found in shallow soil [48], but there is a possibility of a functionally significant proportion of tree fine roots to exist in deeper soil. Further, we made use of the sand in-growth core method to estimate annual fine root production in our plots. This method, just like all other methods of root production measurement [16,28], is imperfect and subject to experimental errors. We contend that the same margin of error would have occurred in all plots, thus facilitating a comparison between the treatments imposed in this study.

## 5. Conclusions

This study makes use of a tree density manipulation experiment set up 30 years ago to consider the carbohydrate investment of regenerating beech trees into resource acquisition. We found that, in the nascent forest stands, although still dominated by beech similar to the original tall forest renewed by the shelterwood management system, the ratio of annual biomass investment into these two compartments is affected by the intensity of mature tree removal and subsequent canopy opening. Our findings highlight the fact that the effects of light availability manipulation in beech forest stands are detectable for decades after intervention. This suggests that the intensity of mature stand harvest in shelterwood systems could be used to optimise carbon and soil nutrient acquisition efficiency.

**Author Contributions:** Conceptualization, B.K. and M.B. (Milan Barna); data curation, M.B. (Michal Bosela) and M.L.; funding acquisition, B.K.; investigation, B.K., M.B. (Milan Barna) and M.B. (Michal Bosela); methodology, B.K. and M.B. (Milan Barna); visualisation, M.B. (Milan Barna) and M.B. (Michal Bosela); supervision, B.K.; writing—original draft, B.K., M.B. (Milan Barna), M.B. (Michal Bosela), and M.L.; writing—review and editing, B.K. and M.L. All authors have read and agreed to the published version of the manuscript.

**Funding:** This research was funded by grant "EVA4.0," No. CZ.02.1.01/0.0/0.0/16_019/0000803 financed by OP RDE, and also by the projects, APVV-18-0086, APVV-19-0387 and APVV-19-0183, from the Slovak Research and Development Agency, as well as by the Grant No. 2/0101/18 supported by the Scientific Grant Agency VEGA of Ministry of Education, Science, Research and Sport of the Slovak Republic and of the Slovak Academy of Sciences.

**Conflicts of Interest:** The authors declare no conflict of interest.

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
