# Peer review of "Biomass Allocation to Resource Acquisition Compartments Is Affected by Tree Density Manipulation in European Beech after Three Decades"

_forests, doi:10.3390/f11090940_

Round 1

Reviewer 1 Report

Dear authors,

Thank you for your interesting input in the field of fine roots/foliage studies. Each piece of research on the topic is very valuable since not a lot is published in this specific field.

I really like the paper, but have nevertheless a few comments, which could improve it (by my opinion).

First, there is possibly an error in abstract - Line 25, the end of sentence – you mention different fine root turnover rates, but both numbers are the same.

Line 91: after 20 there should be unit

Line 122: In Table 2, second column (Silvicultural systems), for NC some of the text is missing

It seems you took soil cores, stored the samples, analysed the roots and used results for root production with ingrowth cores data. Would be interesting to see data from cores, it represents a look into equilibrium state of roots at certain point. From our root studies, we see that between years maximum amount of fine roots in the ground is more or less constant, what changes is ratio between live/dead roots.

In addition, you should add how fine root production was calculated. It seems trivial matter, but in studies like Brunner at al. 2013 it was shown that sometimes different results between studies are simply because of calculation methods.

For table 3 add the unit for production values.

Figure 2 shows average values from all three years per plot?

Figure 3 graph should be bigger as three in a row are quite hard to read. Make them in two rows. Also Table 3 and Figure 3 show the same results. Maybe not neccesary to have both?

Line 232-233 you mention information about ground vegetation. It would be most interesting to see more information on that (not compulsory though ?)

with kind regards

Author Response

editor:

dear authors, It is very interesting research that develops a survey started 30 years ago on density regulation beech forest, located in Western Carpathians, Slovakia. The density regulation was found that has a long-lasting effect on the production of thin roots and foliage in beech. Reading the manuscript, I understood the experimental design, but too many elements are implied. Furthermore, some articles that could explain the applied methodology are not easily found in the databases. Therefore, the experimental design must be expanded to allow the reader to better understand your very interesting study. It is also necessary to give better evidence to the fact that the replicas in your study are there. Make this aspect clear too. Furthermore, it would be good to better explain the part of the statistical analyses. I prepared a list of comment here and added notes to the manuscript.

 List 73 and following Probably, you can give some silvicultural indication from this long survey. Add something in conclusion too. 103-105 I didn't understand. Is the stocking density, the area occupied by canopy trees? or the density in term of tree number? Please give some explication on stand density and on the other parameter you proposed in the table1. Table 1 the reference 19 and 20 are not readable. So please, give some explication in the text. Caption table 1: it must be self-explanatory, please, explicit (even in the manuscript text) what are RD - relative density (%), RI - relative illumination relative throughfall. 114 “All operations are reported for the parent stand only” Which operations? logging operation? Why in CC you have 3 clear cuts? What means cleaning? Please this table 1 is not sufficient for understanding the background of the research, please explain in the text of the manuscript Table 2 Some data in the table are not visible. Please revise the heading of columns, some are not explained. Line 170-180 “2.6. Calculations and statistical analyses” Did you check for normality? Did you check for difference of mean of the parameters of the treatment? 187 figure 2 “Standing stock” You used this term, but you do not give the definition. Number? volume? weight? 216-218 “Finally, plotting the amount of foliage produced in 2013 as a function of standing fine root biomass reveals a linear relationship across the range of density manipulation interventions studied here (Figure 4)”. This is very interesting, please give the statistical parameters of this regression model! I’ll appreciate a carefully discussion of this in Discussion. 286-295 Conclusion Which indication for the manager? You can give some indication, as requested in introduction. 

We would like to thank the editor and both reviewers for an extensive review of the paper and the great number of comments and suggestions. We have corrected many of the issues pointed out and added more detailed explanation where required. As a result, we believe we now have a much better paper. The detailed response to reviewer comments is as follows:

reviewer 1:

Thank you for your interesting input in the field of fine roots/foliage studies. Each piece of research on the topic is very valuable since not a lot is published in this specific field. I really like the paper, but have nevertheless a few comments, which could improve it (by my opinion).

First, there is possibly an error in abstract - Line 25, the end of sentence – you mention different fine root turnover rates, but both numbers are the same.

Line 91: after 20 there should be unit

Line 122: In Table 2, second column (Silvicultural systems), for NC some of the text is missing

Thank you for the suggestions, these points were changed.

It seems you took soil cores, stored the samples, analysed the roots and used results for root production with ingrowth cores data. Would be interesting to see data from cores, it represents a look into equilibrium state of roots at certain point. From our root studies, we see that between years maximum amount of fine roots in the ground is more or less constant, what changes is ratio between live/dead roots.

Unfortunately, dead roots were not measured so we can not answer this query. It is an interesting point though – just from the point of view of understanding root dynamics in these systems.

In addition, you should add how fine root production was calculated. It seems trivial matter, but in studies like Brunner at al. 2013 it was shown that sometimes different results between studies are simply because of calculation methods.

Production was estimated by ingrowth cores, the biomass of fine roots found inside the ingrowth cores was considered as production (starting from zero, i.e. rootless status at the time of installation). 

For table 3 add the unit for production values.

Done

Figure 2 shows average values from all three years per plot?

Yes – this information was added.

Figure 3 graph should be bigger as three in a row are quite hard to read. Make them in two rows. Also Table 3 and Figure 3 show the same results. Maybe not necessary to have both?

Thank you, we have removed Table 3.

Line 232-233 you mention information about ground vegetation. It would be most interesting to see more information on that (not compulsory though).

We have added information: lines 96-98 and citation 19.

[19] Kukla, J., Kontriš, J., Kontrišová, O., Gregor, J., Mihálik, A. 1998. Causes of floristical differentiation of Dentario bulbiferae-Fagetum (Zlatnik 1935) Hartmann 1953 and Carici pilosae-Fagetum Oberd. 1957 Associations. Ekologia Bratislava, 17:177-186.

Reviewer 2 Report

General comments

This paper presents information on annual fine root and litterfall in European beech forests four years after completion of five stand manipulations that included clearcut, uncut control and shelterwoods. There are three primary flaws in the study that prevent the reported results from being sound at this time. First, stand manipulations were not replicated. Therefore, it is unclear if some of the results are real or due to outliers. Second the fine root data from in-growth bags was collected after eight months in the ground. This is probably not enough time to have established a steady state in fine root dynamics. As a result, these results may not represent actual fine root production and turnover responses to the stand manipulations. Also, root samples were frozen and this could have accentuated live roots to appear as dead roots, and added error to the results. Third, there is not enough information provided in the Methods, Figures, and Tables to understand what took place. Each of these should be detailed enough for the reader to fully understand what was done and the figures and tables should be self-explanatory to the reader.  The paper needs a lot of grammatical editing as listed below for the Introduction section.  

Abstract

  1. Line 21: Insert “the” after “…stock in…”
  2. Lines 25-26: 0.8 year-1 is the same, not different.
  3. Line 27: Insert “the” before “…mass basis…”

Introduction

Page1

  1. Line 35: Delete “in the west” and “in the east.”
  2. Line 36: Insert “the country’s” before “forested area” and delete “of the country.”
  3. Line 37: The abbreviation, a.s.l., should be spelled out in its first use with the abbreviation in parentheses.
  4. Paragraph 1: Those unfamiliar with your forest type may not know what you are referring to when you say, “beech.” For example, is all beech, European beech? Is European beech the dominant beech species in your forest? This could be clarified in the first paragraph and then after that, you could simply say, “beech.”
  5. Line 40: Delete “so” after “especially.”
  6. Line 41: Delete “the” after “when” and insert “a” after “by.”
  7. Line 42: Insert comas before and after “thus.”
  8. Line 44: Replace “or” with “and”; delete “light.”

Page 2

  1. Line 47: Throughout the paper, please use the same term. For example, if you use acquisition compartment, then don’t call it acquisitive compartment.
  2. Line 49: Delete “step.”
  3. Line 50: Change to “availabilities” because the plural term is needed.
  4. Lines 51-52: Consider “The key difference between the dynamics of light and soil resource availabilities over time is that changes in available light are robust and changes in soil resources may be subtle.”
  5. Line 54: Change “for” to “to.”
  6. Line 55: Be consistent in using a hyphen in “life-span”; lifespans or lifespan? Insert “the” before “stem”; delete “course”- this is a term to describe roots and all branches would be considered as course; coma after “branches.”
  7. Lines 55-56: Change to “…while lifespans of the stem, branches, and course roots usually correspond to that of the age of the tree, lifespans of the leaves and fine roots range between a couple of months to several years.”; lifespan or lifespans?
  8. Line 57: Insert a semicolon after “ephemeral.”
  9. Lines 58-59: Delete “…cost-effectiveness is eroded by…tree.” and insert “…cost-effectiveness erodes, they senesce, abscise, and are replaced.”
  10. Line 70: Replace “the investment into” with “carbon investments in”; delete “still.”
  11. Line 72: Delete sentence, “For example,…production.”
  12. Line 73: Be more specific than “bring together observations.”
  13. Line 74: Again, use the same term and insert “acquisition” before “biomass.”

Methods

  1. Lines 83-85: Is this spelled correctly? Spell out “Mountains”; End the sentence at “…E)” ad insert “This area is” before “characterized”; coma after “bedrock.”
  2. Lines 85-86: Be specific in this sentence.
  3. Line s 86-87: Use the terms “aspect” for direction and “slope” for percentage slope; not sure what a.s.l. is. Don’t start sentence with a number.
  4. Lines 88-89: Replace “vegetation period” with “growing season.”

Page 3

  1. Section 2.2: Five plots were established on line 99. But, three plots were described as shelterwoods and one plot was described as a clear-cut on lines 100 and 101. What happened to the other plot (control)?
  2. Section 2.2: You describe the treated areas after the first stand manipulation in 1988/89 based on a fraction of an original value of 0.9 for the control plot. But, there is no explanation of what these values are based on (i.e., 0.7, 0.5, 0.3, and 0). A variable that is visual such as basal area is needed.
  3. Section 2.2: Equally, your descriptions of stand manipulations in 2004 is not informative. And Table 1 does not provide the information needed for the reader to visualize stocking and basal area. As a reader, I want to know what the trees per ha and basal area per ha are of each treatment and harvest. Also, you have not explained how you measured relative illumination and relative throughfall before reported these variables in table 1.
  4. In the Introduction, I thought you would be measuring leaf area and fine root biomass in response to different stand manipulations. I did not get the impression that the entire stand would be cut down and then four years later, you would be measuring these variables relative to post-harvest natural reproduction. You need to reorganize your Introduction to preface this effort.

Page 4

  1. Section 2.3: Please be more informative in your methodology. When you describe your tree measurements, please define what qualifies as a tree. For example, what species of tree, what time of year, and what minimum root collar diameter to be counted? How were sprouts from an old root system counted? How were these variables measured and with what equipment? After cessation of growth but before foliage senescence? Must be so if you used foliage to identify tree species.
  2. Section 2.3: Was this just beech or all hardwoods? Table 2 described only beech.
  3. Table 2: What is under “control plot”? Correct you comas and periods in the numbers; Use stems ha-1 and not N ha-1; how did you calculate basal areas of such small stems? Describe how you did this; Use “trace” instead of “neglect”; Define “Beech Prop.” and use a different term without an abbreviation; What is 25-75%?
  4. Section 2.4: It is unclear if both the soil cores and in-growth bags were samples from the four circular areas per subplot in figure 1. If so, more information is needed about how far they were apart from each other.
  5. Line 125: Begin this sentence clarifying which was done- soil core or in-growth bags; Four sets of soil cores per subplot in spring 2013, but there are only four sample circles per subplot in figure 1. More information is needed for the reader to understand what was done.
  6. Line 128: The word, “horizons” is a technical term of soil pedology. Partitioning the soil into 10 cm deep layers does not result in soil horizons, but simply produces four soil subsamples by 10 cm increment of soil depth.
  7. Line 129: Freezing these samples may have been a problem if live root cells were lysed by ice formation. I store samples just above freezing to prevent this from happening.
  8. Line 131: Add the model number and manufacturer of this material in parentheses so that the reader can duplicate the bags.
  9. Line 132: Consider, “…to fit the inside of a metal auger with a 6.5 cm inside diameter and 30 cm length.”
  10. Line 133: It is unclear if the bags were inserted before or after filling them with sand. This write-up must be made clearer without grammatical errors so that the reader can duplicate what you did.
  11. Lines 133-135: As I understand, four in-growth cores were installed per subplot in March 2013 and then after the growing season in November 2013, the in-growth cores were excavated so they were only in the ground for eight months. Then this was repeated in 2014 and 2015. If this is what you did, you need to describe it so. First, you really need to make it clear how long the in-growth cores were in the ground. Second, if this is true and the in-growth cores were only in the ground for eight months, it is unlikely that fine root growth reached an equilibrium that would allow accurate estimation of fine root biomass and turnover. My understanding is that in-growth cores need to be left in the soil for a period of time before normal fine root dynamics is established- that is, a steady state is achieved. Otherwise, your results are variable depending on what’s going on as a result of the coring process itself. For example, cutting a large woody root with the auger is going to create a proliferation of emerging roots. It takes time before this effect fades and a steady state is achieved. In Pinus, this takes about a year. Also, you need to make sure you describe how far away the in-growth cores were from each other and if you backfilled after removing one set of in-growth cores. If they were close together, or they were not backfilled, you could have introduced artifacts into your data.
  12. Line 137: Again, I would be concerned about freezing root samples because this would reduce your ability to tell live from dead roots.
  13. Line 140: There is no description of what the sequence of the sequential cores was other than “four sets”. When were these cores taken? Annually, monthly, bimonthly?
  14. You might want to consider subheadings in Section 2.4 such as “Sequential soil core sampling,” “In-growth bag sampling,” and “Fine root processing.”
  15. Line 138: Use in-growth core or in-growth bag consistently and do not mix terms in the paper.
  16. Line 138: Did you wet sieve the soil, or just pick at it?
  17. You will have to provide more description on live vs dead roots. After freezing, I think live roots would look like dead roots.
  18. Be sure to use the term “fine roots” and not just “roots.”
  19. You need to describe how you differentiated between the fine roots of trees and other plants in the in-growth cores/bags.
  20. Normally, you would calculate ash-free dry weights of fine roots from soil.

Page 5

  1. Correct “little” to “litter.”
  2. I’m not able to evaluate the statistical analyses in this study because I am not sure what the experimental design was. I see the main problem being no replication by stand manipulation treatment.
  3. Describe and cite how you converted littler fall to foliage production.

Results

  1. Line 170: Initiated contrasting development of what? Replace “within each of” with “between.”
  2. Lines 175-177: I don’t understand why you have not also described what was done to the plots in 2004 and 2009. So, beech regenerated better with an overwood and other species regenerated better in the CC. Is this new information? This must have something to do with competition for sunlight.
  3. Table 3: These are means and standard deviations of fine root biomass from the soil cores? At what depth-0-30 cm? Foliage biomass in what units per what area?

Page 6

  1. Figure 2: Standing stock form what- in-growth bags or soil cores? How did you calculate turnover-what method, there are several.
  2. Lines 194-197: Both expected observations; agree with stand density as the driver. But you did not measure inter-annual variation in environmental conditions so why are you saying this? What exactly is this- soil and air temperature, light availability, RH, rainfall?
  3. Line 201: This needs to be described in you methods with citations.
  4. Figure 3: Not root to shoot ratio which is a variable reflecting whole root system to whole shoot weights.
  5. Figure 4 and lines 216-217: Why just 2013? If you are going to do this then you need to include data points from all years.

Discussion

  1. Lines 225-227: I don’t believe the LC foliage and fine root response is different from the other treatments for any other reason than experimental error. Without replication of the stand manipulations, you have no way of knowing if this is real.
  2. It is important in the Discussion to compare your values of fine root biomass and turnover with those from other studies and provide citations to validate the accuracy of your root observations.
  3. Line 251: Low turnover rates are possibly due to low in-growth and absence of steady state. If you waited to excavate bags for a year, your numbers might have been more like those of other studies.
  4. Line 277: There are too many limitations of this study to draw conclusions. It is suggested that the data be repackaged to address another research need. Without replication by stand manipulation, I don’t think the authors can compare the stand manipulations. I also think the authors need to cite other research indicating that a steady state of fine root dynamics was reached by 8 months in beech after in-growth bag installation or these data cannot be used in the paper. The sequential soil cores are good to use but the authors need to describe what the sequence of excavation was. A possible new package of information might be natural regeneration, fine root biomass by depth from soil cores, and foliage production dynamics. Perhaps this represents a void of some fundamental information about European beech forests. However, it seems that information on vegetative competition, or perhaps non-beech fine root biomass, needs to be included as a causal factor because the stand manipulation treatments would have had a big effect on competition between beech seedlings and other vegetation. Rather than ANOVA, correlation analyses of the entire data set might be a valid form of statistical analysis to address some questions emerging from the new data set. But I think the 2013 LC data shown in Figure 4 is an outlier and therefore, the LC plot should be excluded from the data.

Author Response

We would like to thank the editor and both reviewers for an extensive review of the paper and the great number of comments and suggestions. We have corrected many of the issues pointed out and added more detailed explanation where required. As a result, we believe we now have a much better paper. The detailed response to reviewer comments is as follows:

Reviewer 2:

General comments

This paper presents information on annual fine root and litterfall in European beech forests four years after completion of five stand manipulations that included clearcut, uncut control and shelterwoods. There are three primary flaws in the study that prevent the reported results from being sound at this time. First, stand manipulations were not replicated. Therefore, it is unclear if some of the results are real or due to outliers. Second the fine root data from in-growth bags was collected after eight months in the ground. This is probably not enough time to have established a steady state in fine root dynamics. As a result, these results may not represent actual fine root production and turnover responses to the stand manipulations. Also, root samples were frozen and this could have accentuated live roots to appear as dead roots, and added error to the results. Third, there is not enough information provided in the Methods, Figures, and Tables to understand what took place. Each of these should be detailed enough for the reader to fully understand what was done and the figures and tables should be self-explanatory to the reader.  The paper needs a lot of grammatical editing as listed below for the Introduction section.  

Thank you for a superbly thorough review of the paper, the suggestions make the paper much better. We have accepted all suggestions and changed the text accordingly, apart from those with bold highlighted comments below.

Abstract

  1. Line 21: Insert “the” after “…stock in…”
  2. Lines 25-26: 0.8 year-1 is the same, not different.
  3. Line 27: Insert “the” before “…mass basis…”

Introduction

Page1

  1. Line 35: Delete “in the west” and “in the east.”
  2. Line 36: Insert “the country’s” before “forested area” and delete “of the country.”
  3. Line 37: The abbreviation, a.s.l., should be spelled out in its first use with the abbreviation in parentheses.
  4. Paragraph 1: Those unfamiliar with your forest type may not know what you are referring to when you say, “beech.” For example, is all beech, European beech? Is European beech the dominant beech species in your forest? This could be clarified in the first paragraph and then after that, you could simply say, “beech.”
  5. Line 40: Delete “so” after “especially.”
  6. Line 41: Delete “the” after “when” and insert “a” after “by.”
  7. Line 42: Insert comas before and after “thus.”
  8. Line 44: Replace “or” with “and”; delete “light.”

Page 2

  1. Line 47: Throughout the paper, please use the same term. For example, if you use acquisition compartment, then don’t call it acquisitive compartment.
  2. Line 49: Delete “step.”
  3. Line 50: Change to “availabilities” because the plural term is needed.
  4. Lines 51-52: Consider “The key difference between the dynamics of light and soil resource availabilities over time is that changes in available light are robust and changes in soil resources may be subtle.”
  5. Line 54: Change “for” to “to.”
  6. Line 55: Be consistent in using a hyphen in “life-span”; lifespans or lifespan? Insert “the” before “stem”; delete “course”- this is a term to describe roots and all branches would be considered as course; coma after “branches.”
  7. Lines 55-56: Change to “…while lifespans of the stem, branches, and course roots usually correspond to that of the age of the tree, lifespans of the leaves and fine roots range between a couple of months to several years.”; lifespan or lifespans?
  8. Line 57: Insert a semicolon after “ephemeral.”
  9. Lines 58-59: Delete “…cost-effectiveness is eroded by…tree.” and insert “…cost-effectiveness erodes, they senesce, abscise, and are replaced.”
  10. Line 70: Replace “the investment into” with “carbon investments in”; delete “still.”
  11. Line 72: Delete sentence, “For example,…production.”
  12. Line 73: Be more specific than “bring together observations.”
  13. Line 74: Again, use the same term and insert “acquisition” before “biomass.”

Methods

  1. Lines 83-85: Is this spelled correctly? Spell out “Mountains”; End the sentence at “…E)” ad insert “This area is” before “characterized”; coma after “bedrock.”
  2. Lines 85-86: Be specific in this sentence.
  3. Line s 86-87: Use the terms “aspect” for direction and “slope” for percentage slope; not sure what a.s.l. is. Don’t start sentence with a number.
  4. Lines 88-89: Replace “vegetation period” with “growing season.”

Page 3

  1. Section 2.2: Five plots were established on line 99. But, three plots were described as shelterwoods and one plot was described as a clear-cut on lines 100 and 101. What happened to the other plot (control)?
  2. Section 2.2: You describe the treated areas after the first stand manipulation in 1988/89 based on a fraction of an original value of 0.9 for the control plot. But, there is no explanation of what these values are based on (i.e., 0.7, 0.5, 0.3, and 0). A variable that is visual such as basal area is needed.

We have adjusted the table and added information to make it self-explanatory, there are some additional citations that provide background info.

Barna, M., Sedmák, R., Marušák, R. 2010. Response of European beech radial growth to shelterwood cutting. Folia Oecologica, 37 (2), pp. 125-136.

The following text has been added to the paper:

In the shelterwood method the mature stand is removed in a series of two or four cuts.  The early cuts are designed to improve vigor and seed production of the remaining trees while preparing the site for new seedlings.  The final cut is made when a sufficient amount of desirable reproduction. This method provides a partial cover of trees which shelters the new seedlings.  When the shelter becomes a hindrance to the growth of the seedlings, rather than a benefit, it is necessary to remove the remainder of the mature stand (Smith et al 1997).

SMITH,  D.M.,  B.C.  LARSEN,  M.J.  KELTY,ANDP.M.S. ASHTON. 1997.The practice of silvicul-ture: Applied forest ecology, 9th Ed. John Wileyand Sons, New York. 537 p.

Sterba, H. 1987. Estimating potential density from thinning experiments and inventory data. For. Sci., 33: 1022-1034.

Sterba, H. 1998. The precision of species proportion by area when estimated by angle counts and yield tables. Forestry, 71: 25-32.

Table 1. Timing and intensity of forest management operations in experimental forest plots, all interventions were carried out in Feb-Mar prior to the onset of the vegetation season. All operations relate to the mature stand only (red.BA – % reduction of original basal area, BA – basal area after intervention (m2ha-1), RD - relative density (G/Gmax), shelter - shelterwood management system).

  1. Section 2.2: Equally, your descriptions of stand manipulations in 2004 is not informative. And Table 1 does not provide the information needed for the reader to visualize stocking and basal area. As a reader, I want to know what the trees per ha and basal area per ha are of each treatment and harvest. Also, you have not explained how you measured relative illumination and relative throughfall before reported these variables in table 1.

We have added extra information to Table 1 to improve clarity.

  1. In the Introduction, I thought you would be measuring leaf area and fine root biomass in response to different stand manipulations. I did not get the impression that the entire stand would be cut down and then four years later, you would be measuring these variables relative to post-harvest natural reproduction. You need to reorganize your Introduction to preface this effort.

We have strengthened the description of this approach in the intro, thank you.

Page 4

  1. Section 2.3: Please be more informative in your methodology. When you describe your tree measurements, please define what qualifies as a tree. For example, what species of tree, what time of year, and what minimum root collar diameter to be counted? How were sprouts from an old root system counted? How were these variables measured and with what equipment? After cessation of growth but before foliage senescence? Must be so if you used foliage to identify tree species

We have added an explanation to the text. Briefly, all trees taller than 1.3 m were measured for DBH and are included in the statistics. Shorter trees are reported only as number per hectare and were not measured individually (Table 2). All regeneration is from seed, no resprouts were found in these plots.

  1. Section 2.3: Was this just beech or all hardwoods? Table 2 described only beech.

We have added information on other species to the paper (Figure 1 and text)

  1. Table 2: What is under “control plot”? Correct you comas and periods in the numbers; Use stems ha-1 and not N ha-1; how did you calculate basal areas of such small stems? Describe how you did this; Use “trace” instead of “neglect”; Define “Beech Prop.” and use a different term without an abbreviation; What is 25-75%?

Done, basal area calculation and several clarifications are in the text.  

  1. Section 2.4: It is unclear if both the soil cores and in-growth bags were samples from the four circular areas per subplot in figure 1. If so, more information is needed about how far they were apart from each other.

Yes, both types of samples were taken within the circular areas, each repeat cohort was 50 cm away from the previous one.

  1. Line 125: Begin this sentence clarifying which was done- soil core or in-growth bags; Four sets of soil cores per subplot in spring 2013, but there are only four sample circles per subplot in figure 1. More information is needed for the reader to understand what was done.

We have clarified this section.

  1. Line 128: The word, “horizons” is a technical term of soil pedology. Partitioning the soil into 10 cm deep layers does not result in soil horizons, but simply produces four soil subsamples by 10 cm increment of soil depth.
  2. Line 129: Freezing these samples may have been a problem if live root cells were lysed by ice formation. I store samples just above freezing to prevent this from happening.

Yes, this is a valid point. Sample freezing is a standard procedure in root research where immediate sample processing is not possible due to operational reasons. We believe freezing did not affect our ability to classify live / dead roots.

  1. Line 131: Add the model number and manufacturer of this material in parentheses so that the reader can duplicate the bags.

Unfortunately we have no knowledge about that (the material was old stock of unknown origin).

  1. Line 132: Consider, “…to fit the inside of a metal auger with a 6.5 cm inside diameter and 30 cm length.”
  2. Line 133: It is unclear if the bags were inserted before or after filling them with sand. This write-up must be made clearer without grammatical errors so that the reader can duplicate what you did.
  3. Lines 133-135: As I understand, four in-growth cores were installed per subplot in March 2013 and then after the growing season in November 2013, the in-growth cores were excavated so they were only in the ground for eight months. Then this was repeated in 2014 and 2015. If this is what you did, you need to describe it so. First, you really need to make it clear how long the in-growth cores were in the ground. Second, if this is true and the in-growth cores were only in the ground for eight months, it is unlikely that fine root growth reached an equilibrium that would allow accurate estimation of fine root biomass and turnover. My understanding is that in-growth cores need to be left in the soil for a period of time before normal fine root dynamics is established- that is, a steady state is achieved. Otherwise, your results are variable depending on what’s going on as a result of the coring process itself. For example, cutting a large woody root with the auger is going to create a proliferation of emerging roots. It takes time before this effect fades and a steady state is achieved. In Pinus, this takes about a year. Also, you need to make sure you describe how far away the in-growth cores were from each other and if you backfilled after removing one set of in-growth cores. If they were close together, or they were not backfilled, you could have introduced artifacts into your data.

Our aim was to compare annual fine root production in five stands. Root growth during the dormant season is minimal in these forests, we installed each annual series of in-growth cores for a period of 8 months only. Given that no fine root production measurement method is perfect, this gives us the best approximation of annual growth. Since the experimental error we inadvertently introduced is the same in all plots, this gives us a reasonable basis to compare the stands.

  1. Line 137: Again, I would be concerned about freezing root samples because this would reduce your ability to tell live from dead roots.
  2. Line 140: There is no description of what the sequence of the sequential cores was other than “four sets”. When were these cores taken? Annually, monthly, bimonthly?

We did not undertake any sequential coring, we sampled standing fine roof biomass only once.

  1. You might want to consider subheadings in Section 2.4 such as “Sequential soil core sampling,” “In-growth bag sampling,” and “Fine root processing.”
  2. Line 138: Use in-growth core or in-growth bag consistently and do not mix terms in the paper.
  3. Line 138: Did you wet sieve the soil, or just pick at it?

First picked and then wet sieved the soil.

  1. You will have to provide more description on live vs dead roots. After freezing, I think live roots would look like dead roots..

This is a standard description used in a number of papers reporting on fine root investigations. There is a very clear difference between live and dead (for some time) fine roots. Recently abscised roots are difficult to distinguish from live, even without freezing. The experimental error, however, should be the same in all treatments, allowing for comparison.

  1. Be sure to use the term “fine roots” and not just “roots.”

Yes, very good point. Thank you.

  1. You need to describe how you differentiated between the fine roots of trees and other plants in the in-growth cores/bags.
  2. Normally, you would calculate ash-free dry weights of fine roots from soil.

Not quite sure what does the reviewer mean, but in any case we did not incinerate the roots to establish ash content.

Page 5

  1. Correct “little” to “litter.”

Little was the intended term.

  1. I’m not able to evaluate the statistical analyses in this study because I am not sure what the experimental design was. I see the main problem being no replication by stand manipulation treatment.

Indeed, the plots were not replicated and the sub-plots were nested within these. The mixed-model structure was adjusted to accommodate this, resulting in somewhat lower power of the test.

  1. Describe and cite how you converted littler fall to foliage production.

Done.

Results

  1. Line 170: Initiated contrasting development of what? Replace “within each of” with “between.”
  2. Lines 175-177: I don’t understand why you have not also described what was done to the plots in 2004 and 2009. So, beech regenerated better with an overwood and other species regenerated better in the CC. Is this new information? This must have something to do with competition for sunlight.

Yes, there is a difference in regeneration dynamic with relation to species composition. We now cite  Barna, Bosela 2015, Barna 2008 as background info. The additional information on 2004 and 2009 interventions in now in Table 1.

  1. Table 3: These are means and standard deviations of fine root biomass from the soil cores? At what depth-0-30 cm? Foliage biomass in what units per what area? We have added the information

Table 3 was removed

  1. Figure 2: Standing stock form what- in-growth bags or soil cores? How did you calculate turnover-what method, there are several.

Standing stock was measured in 2013 only by soil coring, fine root turnover was then calculated as the ration of production over stack. The turnover calculation is possible only for 2013.

  1. Lines 194-197: Both expected observations; agree with stand density as the driver. But you did not measure inter-annual variation in environmental conditions so why are you saying this? What exactly is this- soil and air temperature, light availability, RH, rainfall?
  2. Line 201: This needs to be described in you methods with citations.
  3. Figure 3: Not root to shoot ratio which is a variable reflecting whole root system to whole shoot weights.
  4. Figure 4 and lines 216-217: Why just 2013? If you are going to do this then you need to include data points from all years.

We cannot show root turnover for other years as standing root mass was measured in 2013 only.

Discussion

  1. Lines 225-227: I don’t believe the LC foliage and fine root response is different from the other treatments for any other reason than experimental error. Without replication of the stand manipulations, you have no way of knowing if this is real.

The statistical design was adjusted for the fact that replicate plots were nested within a single stand. The description of the test has been added.

  1. It is important in the Discussion to compare your values of fine root biomass and turnover with those from other studies and provide citations to validate the accuracy of your root observations.

Yes, these were in the text of the paper, e.g. Bunner et al. 2013, Finer et al. 2007, Neumann 2020.

  1. Line 251: Low turnover rates are possibly due to low in-growth and absence of steady state. If you waited to excavate bags for a year, your numbers might have been more like those of other studies.

Waiting for steady state tends to underestimate fine root turnover, as there is aa increasing possibility of fine root death and disappearance with longer residence time in the soil. Elsewhere we found that the in-growth cores needed to be changed 4 times a year to avoid this problem. Root turnover in the LC plot corresponds with published European data.

  1. Line 277: There are too many limitations of this study to draw conclusions. It is suggested that the data be repackaged to address another research need. Without replication by stand manipulation, I don’t think the authors can compare the stand manipulations.

The statistical analysis was adjusted for this fact, we are confident that we report only significant effects.

I also think the authors need to cite other research indicating that a steady state of fine root dynamics was reached by 8 months in beech after in-growth bag installation or these data cannot be used in the paper.

As pointed out above, we aimed to capture annual fine root production and compare that among the stands. This has been clarified in the paper. Further, when measuring fine root turnover, reaching a steady state within in-growth cores may lead to an underestimate.

The sequential soil cores are good to use but the authors need to describe what the sequence of excavation was.

We did not undertake a sequential coring analysis, standing root biomass was sampled in 2013 only.

A possible new package of information might be natural regeneration, fine root biomass by depth from soil cores, and foliage production dynamics. Perhaps this represents a void of some fundamental information about European beech forests. However, it seems that information on vegetative competition, or perhaps non-beech fine root biomass, needs to be included as a causal factor because the stand manipulation treatments would have had a big effect on competition between beech seedlings and other vegetation.

This has been added to the paper, thank you.

Rather than ANOVA, correlation analyses of the entire data set might be a valid form of statistical analysis to address some questions emerging from the new data set. But I think the 2013 LC data shown in Figure 4 is an outlier and therefore, the LC plot should be excluded from the data.

Based on the analyses of residuals, we found no evidence that the LC plot is an outlier.

Our reactions to the comments inserted in the text of manuscript:

Did you check for normality?

line 279: We used QQ plots to inspect if the residuals of the mixed models follow normal distribution.

Did you checked for differences... ?

This is explained in the lines 166-168.

 "standing stock of fine roots" is always  biomass of fine roots expressed per area unit (alias weight of dry mass expressed per area unit) - we think that this is well-know basic term and does not need to be explained... 

The shadow in the Figure 4. We added explanation in the figure caption.

Round 2

Reviewer 2 Report

A lot of time and effort has gone into this project and it would be great to see it published, but only with conclusions based on sound experimental design and sampling methods. The authors have done a terrific job of addressing comments about the earlier draft of this paper. However, two problems are still present in my opinion, and should be addressed before publication.

First, lack of replication in management treatments is still a problem in the way the data was analyzed. The authors found a very robust difference in fine root and foliage biomass that separates LC from all other management treatments. Yet, there is no way to be sure this is a treatment effect or is due to some other factor that is dissimilar among the treatment plots. For example, one might attribute the LC response to a high fir stocking and the effect of fir competition for water on carbon allocation to beech foliage and fine roots. I realize that the experimental design with nested sub-plots is doable. But, that applies to treatment plots that are the same except for the management applied to them. It is suggested that another type of analysis be considered such as multiple regression with many independent variables measured at each subplot. These variables would include detailed information on competing tree species and ground vegetation (page 9, lines 278-79). I would be cautious making the statement on page 9, lines 307-309. My understanding is you are saying this because there was no difference in fine root in-growth in 8 months and foliage biomass in 2013, 2014, and 2015, or fine root turnover in 2013 among the CC, HC, MC and NC treatments.  But, you still do not have a good reason why the LC values are so different. I think it would be really interesting if you found out why the LC plot is behaving this way... and they are which is obvious from your repeated observations in 2013, 2014, and 2015.

Second, I do agree that 8 months is enough time to determine treatment effects on fine root production even though it is not enough time to reflect standing crop. But, I do not agree that 8 months is enough time to establish a steady state of fine root production in temperate tree species. If the authors want to continue with this, at the very least, they should provide examples and references stating that 8 months is adequate to reach equilibrium. I do not think that fine root turnover can be calculated as the ratio of in-growth core fine root biomass after 8 months and standing crop fine root biomass outside the in-growth cores. The authors indicate that LC fine root turnover was significantly higher than that of the other treatments but was near “normal” for beech. They also indicate fine root turnover for the other 4 treatments was low compared to “normal.” An explanation for this is that the in-growth cores had not yet reached equilibrium with the rooting environment outside the in-growth cores due to root and soil disruption during in-growth core installation. As a result, the numerator in these calculations is too small. I don’t think the fine root turnover information in this paper is strong enough to report.

Author Response

We would like to thank the reviever for valuable comments, the paper is much stronger now as a result. We have improved the flow of the text in places (see attached track changed doc), added more information in methodology, adjusted the conclusions and corrected a number of errors.

Little litter – indeed, litter was the correct term in this instace, thank you for persevering. The error probably originated from a find&replace having gone rogue.

Now, the two outstanding issues:

First, lack of replication in management treatments is still a problem in the way the data was analyzed. The authors found a very robust difference in fine root and foliage biomass that separates LC from all other management treatments. Yet, there is no way to be sure this is a treatment effect or is due to some other factor that is dissimilar among the treatment plots. For example, one might attribute the LC response to a high fir stocking and the effect of fir competition for water on carbon allocation to beech foliage and fine roots. I realize that the experimental design with nested sub-plots is doable. But, that applies to treatment plots that are the same except for the management applied to them. It is suggested that another type of analysis be considered such as multiple regression with many independent variables measured at each subplot. These variables would include detailed information on competing tree species and ground vegetation (page 9, lines 278-79). I would be cautious making the statement on page 9, lines 307-309. My understanding is you are saying this because there was no difference in fine root in-growth in 8 months and foliage biomass in 2013, 2014, and 2015, or fine root turnover in 2013 among the CC, HC, MC and NC treatments.  But, you still do not have a good reason why the LC values are so different. I think it would be really interesting if you found out why the LC plot is behaving this way... and they are which is obvious from your repeated observations in 2013, 2014, and 2015.

We have used simple and multiple regression models to test for the influence of contributing variables such as stand density, beech % or ground vegetation cover. Unfortunately, we do not have a suffcinet numbe of data points to draw a meaningful conclusion. We have highlighted this fact in the text now. As to the pre-existing mixed-model analysis, we have retained this in the text as this is a standard approach to data analysis from this type of experimental setup. Indeed, were we able to travel back in time and start the experiment with today’s knowledge, a ‚proper‘ replication of treatmnets woudl have been enacted.

Second, I do agree that 8 months is enough time to determine treatment effects on fine root production even though it is not enough time to reflect standing crop. But, I do not agree that 8 months is enough time to establish a steady state of fine root production in temperate tree species. If the authors want to continue with this, at the very least, they should provide examples and references stating that 8 months is adequate to reach equilibrium. I do not think that fine root turnover can be calculated as the ratio of in-growth core fine root biomass after 8 months and standing crop fine root biomass outside the in-growth cores. The authors indicate that LC fine root turnover was significantly higher than that of the other treatments but was near “normal” for beech. They also indicate fine root turnover for the other 4 treatments was low compared to “normal.” An explanation for this is that the in-growth cores had not yet reached equilibrium with the rooting environment outside the in-growth cores due to root and soil disruption during in-growth core installation. As a result, the numerator in these calculations is too small. I don’t think the fine root turnover information in this paper is strong enough to report.

 Root turnover – we have explained the nature and the limitations of our approach and highligted that the fine root turnover data are for comparison between treatments only. A clear definition of how turover was calculated has been added. The steady state discussion is perhaps a topic that merits a separate paper. If one defines fine root turnover as the ratio between annual production and standing stock, it is the standing stock that should be in a steady state. For the root mass in the inrowth core to reach a steady state, a certain amoount of root diaback need to accur – leading to an underestimate of production. Granted, too frequent replacement if ingrowth cores leads to compounding of artifacts so an optimum period suitable for a given ecosystem must be found. We feel that a single growing season is sufficient in our systems, but as already stated – this may merit its own study.

Authors

24 Aug 2020